# Effects of Nitrogen on Photosynthetic Productivity and Yield Quality of Wheat (*Triticum aestivum* L.)

Hafeez Noor [ID], Zhouzuo Yan, Peijie Sun, Limin Zhang, Pengcheng Ding, Linghong Li, Aixia Ren, Min Sun * and Zhiqiang Gao

College of Agriculture, Shanxi Agriculture University, Jinzhong 030801, China;
hafeeznoorbaloch@gmail.com (H.N.); noorbaloch2@yahoo.com (Z.Y.); doctoornoor@gmail.com (P.S.);
wenliin597@gmail.com (L.Z.); shahbazkhanbaloch30@gmail.com (P.D.); litingliang77@gmail.com (L.L.);
aixiaren9@gmail.com (A.R.); gaozhiqiang67@gmail.com (Z.G.)
* Correspondence: sunmin@sxau.edu.cn

**Abstract:** This study aims to understand the influence of chlorophyll fluorescence parameters on the yield of winter wheat in some areas of China. Nitrogen (N) application is believed to improve photosynthesis in flag leaf, which ultimately increases the final yield. The experiment was conducted in the wheat experimental base of Shanxi Agricultural University in Taigu, Shanxi Province, China; before sowing, four N application rates were set—N0, N120, N150, and N210 kg ha$^{-1}$ of the Yunhan-20410 variety from 2019 to 2022. The results from different parameters of research showed that the organic manure partial substitution for chemical fertilizer increased post-anthesis N uptake by 16.4 and 81.4%, thus increasing the post-anthesis photosynthetic capacity and delaying leaf senescence. N150 treatment can improve dry matter (DM) accumulation, thus promoting the increase of the yield. The maximum net photosynthesis $P_N$ value of the booting stage and flowering stage indicated that nitrogen application could significantly improve the photosynthetic rate of wheat leaves, among which medium nitrogen treatment had the most significant promoting effect. The single-photon avalanche diode (SPAD) value of the leaf of wheat in each treatment increased rapidly in a small range from the jointing stage to the booting stage, respectively. The grain yield under N fertilizer partial substitution for N fertilizer treatment increased by 23%. According to the different significance test, the effects of nitrogen application rate on net photosynthesis $P_N$ of winter wheat were extremely significant at all growth stages, indicating that changing the population distribution mode and nitrogen level could effectively improve leaf photosynthetic performance and that N150 level was the best.

**Keywords:** chlorophyll fluorescence; dry matter; photosynthetic characteristics; wheat; yield components

## 1. Introduction

Rainfed agriculture in Asian and Pacific regions accounts for approximately 70% of the region's arable land, and 60–80% of global food supply comes from these rainfed lands [1]. The successful cultivation and promotion of superior varieties are the basis for the continuous increase of wheat yield and grain quality [2]. Under drought conditions, the plant stomata rates close, due to which the plant's absorption of $CO_2$ reduces, and the rate of photosynthesis growth and yield of the plant is affected badly [3]. High planting density has been adopted in sprout production systems, which improves photosynthesis and influences plant height, architecture, and synthesis of chlorophyll [4]. Leaf senescence comprises a series of biochemical and physiological events from the fully expanded state until death. The leaf duration after full expansion depends strongly on the water conditions and crop species; some researchers have reported that the post-anthesis senescence in cereals affects the whole plant, with organs closest to the developing grains' flag leaves generally senescing last [5]. Nitrogen is the material basis for the synthesis of wheat protein.

Within a certain range, with the increase of nitrogen application level, the protein content of wheat grains increases, but excessive or insufficient nitrogen application will reduce the transport of accumulated nitrogen before anthesis to grains and affect the protein content of grains [6]. The photosynthetic duration is closely related to leaf aging [7]. The premature photosynthetic nitrogen transport in wheat results in premature canopy leaf senescence and grain yield reduction [8]. The green leaf area has significant effects on the grain yield, and shaded leaves become senescent earlier compared to unshaded leaves [9]. Net photosynthesis ($P_N$), stomatal conductance ($gs$), and transpiration rate ($E$), as well as increasing the substomatal $CO_2$ concentration ($C_i$) in wheat, increase the N content, considered as the value of wheat stomatal restriction in different plant populations [10]. However, the process of nitrogen transport from wheat organs to grains is extremely complex in physiology. Greenhouse-scale studies have no way to determine which photosynthetic nitrogen and stored nitrogen transport starts first, or even synchronously, but the transport ratio of photosynthetic nitrogen and stored nitrogen can be calculated [11].

The root-cutting maintained higher photosynthesis, but significantly reduced transpiration and stomatal conductance [12,13]. Maintaining higher photosynthesis and reducing transpiration water loss are important for improving water use efficiency. The grain-filling process mainly depends on three aspects of material supply, including the direct transport of photosynthesis from leaves and stem sheaths to grains during grain filling [14]. Therefore, dry matter accumulation and transport during the wheat grain-filling stage are very important processes for yield formation and water use efficiency. However, there was little information on dry matter (DM) accumulation and transport during the wheat-filling stage, the effects of fertilizer on wheat yield, and water use efficiency. In general, the N application of chemical fertilizers can increase biomass accumulation and thus yield. Nevertheless, excessive biomass accumulation leads to excessive canopy leaf area, resulting in excessive increase of transpiration and water loss and ultimately excessive consumption of soil water [15]. N fertilizer utilization not only improved soil water retention capacity by improving soil aggregate components (>0.25 mm), but also significantly increased soil nutrient (N, P, K, and organic matter) contents in the growth period and ultimately significantly increased crop yield and water use efficiency [16].

Nitrogen fertilizer is a major component of proteins and nucleic acids, and its application regulates plant growth [17]. Nitrogen fertilizer inhibits the efficiency of photosynthesis and irradiation to reduce grain yield, while the optimal concentration of nitrogen application can increase grain yield [18,19]. A previous study suggested that N fertilizer has a positive correlation with the photosynthetic efficiency [20]. The objectives of this study were to (i) to study the changes in the photosynthetic stage, jointing stage, and booting stage, and (ii) to study the response of physiological characteristics of nitrogen use, light use, leaf senescence, yield formation, and quality of wheat. Our study has important implications for assessing the nitrogen effects of physiological characteristics as well as for the sustainable development of greenhouses.

## 2. Materials and Methods

The experiment was conducted in the wheat experimental base of Shanxi agricultural university in Taigu, Shanxi Province, China (E112°34′ E, N37°25′ N), which belongs to the temperate continental climate zone, with an average annual temperature of 10.4 °C. The area is separated by concrete walls with a thickness of 20 cm, and an insulation layer of 10 cm was added to the outer walls.

### 2.1. Experimental Design and Crop Management

The experiment was arranged in a two factors split-plot design with the four N application rates N0, N120, N150, and N210 kg ha$^{-1}$ variety Yunhan-20410 from 2019 to 2022. There was a widely planted wheat N application rate of 97.5 kg ha$^{-1}$ in early October each year. The plot size was 2 m × 4 m = 8 m$^2$. Before sowing, 150 kg $P_2O_5$ ha$^{-1}$ and 150 kg $K_2O$ ha$^{-1}$ were evenly applied to the plots. The nitrogen fertilizer was applied to the base

fertilizer. The seeds were sown on 9 November, planted in a manual with row spacing of 20 cm and a sowing quantity of 225 kg ha$^{-1}$. Irrigation was applied using a drip irrigation system, with the application of 50 mm each time as measured with a water meter. All plants were harvested on 2 June. During the whole growing season, weeds were well controlled by hand (Figure 1A,B).

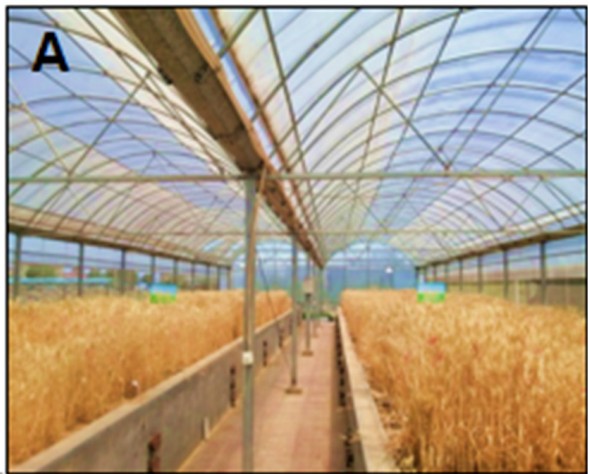
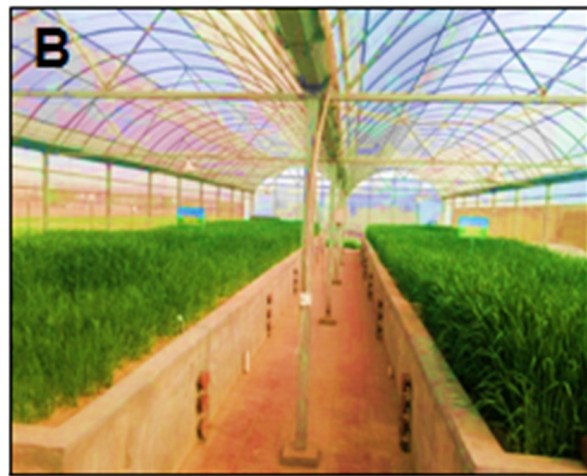

**Figure 1.** Greenhouse at experimental site of Shanxi Agricultural University, Taigu, Shanxi, China. (**A**) Maturity stage of greenhouse winter wheat. (**B**) Jointing stage of greenhouse winter wheat.

### 2.1.1. Photosynthetic Characteristics

Photosynthetic rate $P_N$, in the different stages of winter wheat (Jointing, Booting, Flowering, Filling), in sunny and cloudless weather. The field atmospheric $CO_2$ concentration (about 380 µmol L$^{-1}$) and artificial light source were used to set the optical quantum density. The photon flux density (PFD) was 1200 µmol m$^{-2}$ s$^{-1}$, the airflow rate was 500 µmol s$^{-1}$, and the leaf chamber temperature was 27.7 °C. Nine flag leaves with consistent growth and a similar light direction were selected for each treatment. The diurnal variation of the photosynthetic rate was measured every 4 h from 8:00 to 12:00 am under sunny weather.

### 2.1.2. Photosynthetic Pigment and Chlorophyll Fluorescence Parameters

The chlorophyll fluorescence characteristics of wheat flag leaves were measured at the jointing stage, booting stage, flowering stage, and filling stage. A portable modulated chlorophyll fluorimeter (PAM-2500, Heinz Walz GmbH, Pfullingen, Germany) was used to determine the chlorophyll fluorescence induction kinetics parameters of fully expanded wheat flag leaves with consistent growth and light exposure from 8:00 am to 12:00 am every day.

The wheat was kept in the dark for at least 20 min. After dark treatment, Chl fluorescence induction curves were measured using plant efficiency analyzer (PEA) with red irradiance of 3000 µmol (photon) m$^{-2}$ s$^{-1}$. Measurements were performed at 12:00 and 17:00 h on the same day. The SPAD-502 measurements were conducted in the field between 10:00 and 16:00 h. The adaxial side of the leaves was always placed toward the emitting window of the instrument, and major veins were avoided. The area of the punched wheat leaf material was determined using an area meter (Delta-T Devices Ltd., Cambridge, UK), while two circular 1.0 cm diameter leaf discs from one side on the midrib were punched for wheat. The spatial distribution of the punched leaf material matched the distribution of the single-photon avalanche diode SPAD-502.

### 2.1.3. Grain Yield and Yield Component

At maturity, 20 plants from each plot were randomly sampled from the inner rows to determine yield components such as ear number, grain number per ear, and 1000-grain

weight. Grain yield was obtained by harvesting all plants in the plot. After the plants were mechanically dispelled and the grains were air-dried, the moisture content of the grains was measured to 12.5%.

2.1.4. Statistical Analysis

The data of photosynthesis physiological and winter wheat growth yield were processed and statistically analyzed through Microsoft Excel 2010 and Sigma plot 14.0 software to process data and draw graphs, and DPS7.5 was used for statistical analysis. A two-way ANOVA was used to study the main influence and interaction of variable fluorescence types on yield. When there was a significant interaction effect between photosynthesis physiological, SPAD, and yield, the least significant difference (LSD) method was used for variance analysis and independent $T$ test, and the significance level was set to $\alpha = 0.05$. Differences were considered statistically significant when $p \leq 0.05$.

**3. Results**

*3.1. Photosynthetic Characteristics*

Net photosynthesis ($P_N$) reached the maximum value at the flowering stage and began to decline after the flowering stage (Table 1). The maximum net photosynthesis $P_N$ value of the booting stage and flowering stage was 20.06, which was increased by 15.37%, 6.97%, and 5.88% compared to of nitrogen rate, respectively, indicating that nitrogen application could significantly improve the photosynthetic rate of wheat leaves, among which medium nitrogen treatment had the most significant promoting effect. However, the high nitrogen application rate did not improve the net photosynthetic rate of wheat leaves but inhibited it to some extent. The appropriate demographic structure was useful for accumulating net photosynthesis, and N150 treatment significantly enhanced the net photosynthesis $P_N$ increase in winter wheat flag leaves, whereas excessive nitrogen use had a particular inhibitory effect. Meanwhile, there was a significant difference between them in N0 and N210 treatment and a significant difference between the treatments in the flowering stage. According to the different significance test, the effects of nitrogen application rate on net photosynthesis $P_N$ of winter wheat were extremely significant at all growth stages, indicating that changing population distribution mode and nitrogen level could effectively improve leaf photosynthetic performance and N150 treatment was the best.

Effects of different N application comparison of stomatal conductance $g_s$, are presented in Table 2. In the flowering stage, N150 was the highest, reaching 0.45, followed by flowering stage and filling stage, N210, and 10–20 d N150 treatment. The stomatal conductance $g_s$ value was highest in the N150 treatment compared to other treatments, and there was a significant difference between N0 and N120 treatments. There was a significant difference between treatments N120 and N150. Nevertheless, there was a significant difference between N210 treatments. The N150 treatment had a very significant effect on stomatal conductance $g_s$ at each growth stage, but only N150 had a very significant interaction at the booting stage and a significant interaction at the filling stage.

Effects of different nitrogen applications are presented in Table 3. The transpiration rate ($E$) mmol ($H_2O$) m$^{-2}$ s$^{-1}$ of each treatment overall trend of first increasing, and then decreasing. Transpiration rate leaves were increased slowly from the jointing stage to the booting stage, and the average value increased by 78%, reaching the maximum value at the flowering stage. The average growth from booting to flowering was 4.30 mmol m$^{-2}$ s$^{-1}$. The transpiration rate value of N120 and N150 treatment at the booting stage was greater than that of N0 and N210 treatment, and there was no significant difference between N0 and N120 treatment. The transpiration rate value of booting stage and flowering stage was significantly higher than that of nitrogen treatment at the filling stage and reached the maximum value at N150 and N210 treatment, except that nitrogen application N0 treatment had no significant effects on transpiration rate mmol ($H_2O$) m$^{-2}$ s$^{-1}$ filling stage of the nitrogen application. The interaction effects of nitrogen application rate on transpiration

rate $E$ were the largest at the flowering stage, so optimizing the combination of nitrogen application rate could enhance the leaf function of wheat at the late stage.

**Table 1.** Effects of nitrogen application comparison of net photosynthesis $P_N$ of winter wheat.

| Year | N Application (kg ha$^{-1}$) | Growth Stage | | | |
|---|---|---|---|---|---|
| | | Jointing Stage | Booting Stage | Flowering Stage | Filling Stage |
| 2019–2020 | N0 | 18.0 c | 28.0 c | 31.0 b | 32.0 b |
| | N120 | 21.0 c | 32.0 b | 34.0 b | 32.0 b |
| | N150 | 25.0 a | 34.0 a | 41.0 a | 35.0 a |
| | N210 | 22.0 b | 34.0 a | 40.0 a | 31.0 b |
| | Mean | 0.22 | 0.32 | 0.37 | 0.32 |
| 2020–2021 | N0 | 22.0 d | 30.0 b | 37.0 b | 31.0 b |
| | N120 | 25.0 c | 32.0 b | 40.0 b | 33.0 b |
| | N150 | 28.0 a | 36.0 a | 44.0 a | 37.0 a |
| | N210 | 27.0 b | 35.0 a | 42.0 ab | 32.0 b |
| | Mean | 0.26 | 0.33 | 0.41 | 0.33 |
| 2021–2022 | N0 | 17.0 c | 27.0 c | 30.0 b | 31.0 b |
| | N120 | 20.0 c | 31.0 b | 33.0 b | 31.0 b |
| | N150 | 24.0 a | 33.0 a | 40.0 a | 34.0 a |
| | N210 | 21.0 b | 33.0 a | 39.0 a | 30.0 b |
| | Mean | 0.22 | 0.32 | 0.37 | 0.32 |
| ANOVA | | | | | |
| Y | | ** | ** | ** | ** |
| N | | ** | ** | ** | ** |
| Y × N | | ns | ** | ns | * |

Note: Within a column for each nitrogen rate, means followed by different lower-case letters are significantly different according to Tukey's HSD test (0.05). * and **, significant at 0.01 and 0.05 probability levels, respectively; ns, indicates not significant at 0.05 probability level.

The influence of different nitrogen applications in the substomatal $CO_2$ concentration $C_i$ of winter wheat flag leaves is presented in Table 4. The maximum substomatal $CO_2$ concentration $C_i$ value of days was 505.28 substomatal $CO_2$ concentration $C_i$ (μmol ($CO_2$) mol$^{-1}$). With the increase in nitrogen application rate, the substomatal $CO_2$ concentration $C_i$ value decreased significantly. The substomatal $CO_2$ concentration $C_i$ value of the flowering stage, the filling stage N120 treatment at the filling stage, and the N120 treatment at the booting stage might be due to the high nitrogen level. Environmental stress on the flag leaf causes the stomata to close on the wheat flag leaf, resulting in an increase in substomatal $CO_2$ concentration $C_i$. The N120 treatment at all stages of growth had significant effects on the substomatal $CO_2$ concentration $C_i$ winter wheat flag leaf, and the interaction between these two interactions was extremely significant at all stages of growth except the booting stage.

### 3.2. Effects of Nitrogen Application on SPAD Leaf of Winter Wheat

The SPAD values in leaves of winter wheat are presented in Table 5. The SPAD value of the leaf of wheat in each treatment increased rapidly in a small range from the jointing stage to the booting stage, with an average growth rate of 13.2%. The SPAD value of the N150 and N210 treatment was the highest. The SPAD value of N150 treatment was 48.81, which was 1.04 times N0 and 1.02 times N120. The amount of SPAD of leaves under different conditions varied according to the amount of nitrogen consumption on different days. N0 treatment had no significant difference with N150 except that there was a significant difference between N150 and N210 treatment at the booting stage.

**Table 2.** Effects of different N application comparison of stomatal conductance $g_s$ of winter wheat.

| Years | N Application (kg ha$^{-1}$) | Growth Stage | | | |
|---|---|---|---|---|---|
| | | Jointing Stage | Booting Stage | Flowering Stage | Filling Stage |
| 2019–2020 | N0 | 0.19 c | 0.27 c | 0.32 b | 0.30 b |
| | N120 | 0.20 c | 0.31 b | 0.35 b | 0.31 b |
| | N150 | 0.26 a | 0.35 a | 0.42 a | 0.36 a |
| | N210 | 0.22 b | 0.34 a | 0.40 a | 0.31 b |
| | Mean | 0.22 | 0.32 | 0.37 | 0.32 |
| 2020–2021 | N0 | 0.22 d | 0.30 b | 0.37 b | 0.31 b |
| | N120 | 0.25 c | 0.32 b | 0.40 b | 0.33 b |
| | N150 | 0.29 a | 0.37 a | 0.45 a | 0.38 a |
| | N210 | 0.27 b | 0.35 a | 0.42 ab | 0.32 b |
| | Mean | 0.26 | 0.33 | 0.41 | 0.33 |
| 2021–2022 | N0 | 0.18 b | 0.28 b | 0.29 c | 0.25 c |
| | N120 | 0.19 b | 0.30 ab | 0.32 b | 0.26 c |
| | N150 | 0.22 a | 0.31 a | 0.38 a | 0.33 a |
| | N210 | 0.19 b | 0.28 b | 0.37 ab | 0.31 b |
| | Mean | 0.19 | 0.29 | 0.34 | 0.29 |
| ANOVA | | | | | |
| Y | | ** | ** | ** | ** |
| N | | ** | ** | ** | ** |
| Y×N | | ns | ** | ns | * |

Note: Within a column for each nitrogen rate, means followed by different lower-case letters are significantly different according to Tukey's HSD test (0.05). * and **, significant at 0.01 and 0.05 probability levels, respectively; ns, indicates not significant at 0.05 probability level.

**Table 3.** Effects of nitrogen application on transpiration rate *E* of winter wheat.

| Year | N Application (kg ha$^{-1}$) | Growth Stage | | | |
|---|---|---|---|---|---|
| | | Jointing Stage | Booting Stage | Flowering Stage | Filling Stage |
| 2019–2020 | N0 | 1.64 ab | 2.00 c | 5.31 b | 4.63 b |
| | N120 | 1.96 a | 2.97 b | 5.56 b | 5.20 ab |
| | N150 | 1.91 a | 3.15 a | 8.07 a | 5.04 ab |
| | N210 | 1.55 b | 2.94 b | 5.96 b | 5.60 a |
| | Mean | 1.76 | 2.77 | 6.22 | 5.12 |
| 2020–2021 | N0 | 2.05 b | 2.30 b | 6.28 c | 5.85 a |
| | N120 | 2.07 b | 2.72 a | 7.52 ab | 5.91 a |
| | N150 | 2.30 ab | 2.64 a | 8.37 a | 6.26 a |
| | N210 | 1.52 c | 2.42 ab | 7.30 b | 6.00 a |
| | Mean | 2.12 | 2.52 | 7.37 | 6.01 |
| 2021–2022 | N0 | 1.78 c | 2.77 b | 5.85 a | 5.01 c |
| | N120 | 2.35 a | 3.11 a | 5.91 a | 6.00 a |
| | N150 | 2.23 b | 3.13 a | 6.26 a | 5.49 b |
| | N210 | 1.17 d | 2.27 c | 6.00 a | 4.80 d |
| | Mean | 1.88 | 2.82 | 6.01 | 5.33 |
| ANOVA | | | | | |
| Y | | ** | ** | ** | ** |
| N | | ** | ** | ** | ns |
| Y × N | | ** | ** | ** | * |

Note: Within a column for each nitrogen rate, means followed by different lower-case letters are significantly different according to Tukey's HSD test (0.05). * and **, significant at 0.01 and 0.05 probability levels, respectively; ns, indicates not significant at 0.05 probability level.

**Table 4.** Effects of different nitrogen application on substomatal $CO_2$ concentration $C_i$ between different nitrogen rates in winter wheat.

| Year | N Application (kg ha$^{-1}$) | Growth Stage | | | |
|---|---|---|---|---|---|
| | | Jointing Stage | Booting Stage | Flowering Stage | Filling Stage |
| 2019–2020 | N0 | 309.00 b | 310.00 c | 456.12 a | 234.00 b |
| | N120 | 327.33 a | 387.67 a | 520.14 a | 242.17 a |
| | N150 | 312.67 b | 354.00 ab | 500.20 a | 221.74 c |
| | N210 | 298.00 c | 332.00 bc | 494.16 a | 252.04 a |
| | Mean | 311.75 | 350.56 | 492.66 | 229.99 |
| 2020–2021 | N0 | 313.67 b | 300.00 b | 409.03 b | 212.00 ab |
| | N120 | 317.67 a | 403.00 a | 528.33 a | 237.53 a |
| | N150 | 305.67 c | 380.00 a | 513.00 a | 231.32 ab |
| | N210 | 307.00 c | 321.33 b | 442.00 b | 198.46 b |
| | Mean | 311.00 | 351.08 | 473.09 | 219.83 |
| 2021–2022 | N0 | 315.33 a | 383.67 b | 400.08 c | 207.96 ab |
| | N120 | 283.33 c | 476.00 a | 558.04 a | 238.46 a |
| | N150 | 308.67 b | 404.00 b | 473.00 b | 209.64 ab |
| | N210 | 306.67 b | 407.67 b | 590.00 a | 183.51 b |
| | Mean | 303.50 | 417.84 | 505.28 | 209.89 |
| ANOVA | | | | | |
| | Y | ** | ** | * | * |
| | N | ** | ** | ** | ** |
| | Y × N | ** | ns | ** | ** |

Note: Within a column for each nitrogen rate, means followed by different lower-case letters are significantly different according to Tukey's HSD test (0.05). * and **, significant at 0.01 and 0.05 probability levels, respectively; ns, indicates not significant at 0.05 probability level.

**Table 5.** Effects of nitrogen application on SPAD value of flag leaf of winter wheat.

| Year | N Application (kg ha$^{-1}$) | Growth Stage | | | | |
|---|---|---|---|---|---|---|
| | | Jointing Stage | Booting Stage | Flowering Stage | Filling Stage | Dough Stage |
| 2019–2020 | N0 | 45.64 b | 51.40 b | 51.77 a | 45.67 a | 41.22 a |
| | N120 | 45.75 ab | 52.34 a | 53.01 a | 46.07 a | 41.84 a |
| | N150 | 46.25 a | 52.73 a | 53.19 a | 47.27 a | 43.70 a |
| | N210 | 46.29 a | 52.79 a | 53.34 a | 47.57 a | 43.24 a |
| | Mean | 45.98 | 52.32 | 52.83 | 46.65 | 42.50 |
| 2020–2021 | N0 | 46.38 a | 52.79 c | 53.1 a | 46.37 a | 42.6 a |
| | N120 | 46.19 a | 53.28 b | 54.72 a | 47.97 a | 44.80 a |
| | N150 | 47.47 a | 54.12 a | 55.75 a | 49.47 a | 45.67 a |
| | N210 | 47.14 a | 53.17 bc | 55.62 a | 47.77 a | 42.17 a |
| | Mean | 46.79 | 53.34 | 54.80 | 47.90 | 43.81 |
| 2021–2022 | N0 | 43.88 b | 49.53 c | 50.97 b | 43.65 a | 40.67 c |
| | N120 | 45.86 a | 49.84 c | 51.37 b | 45.64 a | 42.30 a |
| | N150 | 45.3 a | 51.5 a | 52.77 a | 45.67 a | 41.34 b |
| | N210 | 45.07 a | 50.54 b | 51.77 ab | 45.07 a | 41.17 b |
| | Mean | 45.03 | 50.35 | 51.72 | 45.01 | 41.37 |
| ANOVA | | | | | | |
| | Y | * | ** | * | * | ns |
| | N | ns | ** | * | ns | ns |
| | Y × N | ns | * | ns | ns | ns |

Note: Within a column for each nitrogen rate, means followed by different lower-case letters are significantly different according to Tukey's HSD test (0.05). * and **, significant at 0.01 and 0.05 probability levels, respectively; ns, indicates not significant at 0.05 probability level.

### 3.3. Effects Structure Change Post-Anthesis Nitrogen Use Efficiency of Vegetative Organs in Winter Wheat

The post-anthesis transshipment amount of nitrogen in all nutrient organs and increases of nitrogen application rate are presented in Table S1 (Supplement). The average nitrogen transshipment of N0, N120, N150, and N210 treatment was 62.73 kg ha$^{-1}$, 73.98 kg ha$^{-1}$, 94.89 kg ha$^{-1}$, and 84.15 kg ha$^{-1}$, respectively. N120, N150, and N210 treatment increased by 17.94%, 51.27%, and 34.16%, respectively. The nitrogen application significantly promoted nitrogen transshipment volume of vegetative organs and nitrogen transshipment contribution rate of vegetative organs to grains. N150 treatment significantly promoted nitrogen transshipment volume of vegetative organs, but there was no significant difference in the nitrogen contribution rate of vegetative organs to grains among the three irrigations.

The agronomic use efficiency and nitrogen use efficiency increased firstly and then decreased, while the nitrogen productivity decreased with the increase of nitrogen application rate (Table S2, Supplement). The maximum value of agricultural nitrogen use efficiency and nitrogen use efficiency for different nitrogen application levels was obtained in N150 treatment, which was as follows: 20.81 kg ha$^{-1}$ and 26.19% were 1.28, 1.90 times, and 1.43, 1.75 times of other N application levels, respectively, suggesting that low or excessive N application would reduce the use efficiency of N fertilizer. For N150 treatment, N use efficiency was 21.71 kg ha$^{-1}$, followed by 21.57, N150 treatment. The effects of different irrigation and N application on N agronomic use efficiency and N partial productivity were extremely significant. The effects of nitrogen application, and the interaction of N150 treatment on the nitrogen use efficiency of winter wheat were extremely significant.

### 3.4. Effects of Nitrogen Application on Grain Protein

Nitrogen application amount had significant and extremely significant effects on clear, pellet, alcohol solubility glutenin content, gluten to alcohol ratio, protein content, and yield. The interaction of nitrogen application amount year, nitrogen application amount had extremely significant alcohol solubility, gluten content, glutenin to alcohol ratio, protein content, and yield (Table 6). The contents of clear, spherical, alcohol-soluble, and glutenin in wheat grains were not significantly different, but the protein yield was significantly increased by 13.4–16.3%. The increase of N210 treatment, the contents of clear, pellet, alcohol solubility, gluten, gluten to alcohol ratio, and protein showed an increasing trend. The increase of N150 treatment protein yield showed an increasing trend.

### 3.5. Dry Matter Accumulation and Distribution Characteristics

Dry weight at maturity, dry weight at anthesis, and dry matter (DM) accumulation after anthesis were significantly affected by year, nitrogen application amount, and the interaction and had extremely significant effects on dry weight at maturity, dry weight at anthesis, and DM accumulation after anthesis (Table 7). Dry weight compared to maturity stage was increased by 11.8–15.6%, dry weight at flowering stage was increased by 8.4–12.9%, and dry matter accumulation after flowering was increased by 18.2–25.6%, but the harvest index had no significant difference. The increase of nitrogen application rate N210 treatment the harvest index showed a decreasing trend. The increase of nitrogen application rate N150 treatment dry weight at maturity stage, dry weight at flowering stage, and dry matter accumulation after flowering showed a significant increasing trend. Dry weight at maturity stage, dry weight at flowering stage, and DM accumulation after flowering showed an increasing trend with the continuous increase of nitrogen application rate 150–210 treatment, while dry weight at maturity stage, dry weight at flowering stage, and DM accumulation after flowering showed a decreasing trend under N210. N150 treatment can improve DM accumulation, thus promoting the increase of yield.

**Table 6.** Effects of nitrogen application on grain protein and component contents at maturity of wheat.

| Year | N Application (kg ha$^{-1}$) | Albumin (%) | Gliadin (%) | Glutenin (%) | Glu/Gli | Protein (%) |
|---|---|---|---|---|---|---|
| 2019–2020 | N0 | 1.21 b | 3.78 b | 3.63 b | 0.96 b | 11.69 b |
| | N120 | 1.49 a | 4.16 a | 4.18 a | 1.00 ab | 13.13 a |
| | N150 | 1.55 a | 4.15 a | 4.25 a | 1.02 ab | 13.55 a |
| | N210 | 1.51 a | 4.15 a | 4.22 a | 1.02 a | 13.62 a |
| | Mean | 1.44 | 4.06 | 4.07 | 1.00 | 13.00 |
| 2020–2021 | N0 | 1.22 b | 3.78 c | 3.68 c | 0.97 b | 11.63 c |
| | N120 | 1.51 a | 4.16 b | 4.05 b | 0.97 b | 13.05 b |
| | N150 | 1.59 a | 4.19 b | 4.06 b | 0.97 b | 13.25 b |
| | N210 | 1.52 a | 4.35 a | 4.39 a | 1.01 a | 14.15 a |
| | Mean | 1.46 | 4.12 | 4.05 | 0.98 | 13.02 |
| 2021–2022 | N0 | 1.37 b | 3.91 c | 3.86 c | 0.99 b | 11.71 c |
| | N120 | 1.64 a | 4.23 b | 4.21 b | 0.99 b | 13.26 b |
| | N150 | 1.65 a | 4.26 b | 4.22 b | 0.99 b | 13.42 b |
| | N210 | 1.68 a | 4.55 a | 4.59 a | 1.01 a | 13.99 a |
| | Mean | 1.59 | 4.24 | 4.22 | 1.00 | 13.09 |
| ANOVA | | | | | | |
| Y | | ** | ** | ** | * | ** |
| N | | ns | ns | ns | ns | ns |
| Y × N | | ** | ** | ** | ** | ** |

Note: Within a column for each nitrogen rate, means followed by different lower-case letters are significantly different according to Tukey's HSD test (0.05). * and **, significant at 0.01 and 0.05 probability levels, respectively; ns, indicates not significant at 0.05 probability level.

**Table 7.** Effects of nitrogen application on dry matter accumulation and distribution.

| Year | N Application (kg ha$^{-1}$) | TDW (kg ha$^{-1}$) | HI (%) | TDW$_{as}$ (kg ha$^{-1}$) | TDW$_{post}$ (kg ha$^{-1}$) |
|---|---|---|---|---|---|
| 2019–2020 | N0 | 10,619.0 d | 46.0 a | 6829.8 d | 3789.2 c |
| | N120 | 18,591.7 c | 42.6 b | 11,872.6 c | 6719.1 b |
| | N150 | 19,628.8 b | 42.1 b | 12,400.9 b | 7227.9 a |
| | N210 | 20,178.1 a | 41.9 b | 12,927.9 a | 7250.2 a |
| | Mean | 17,254.4 | 43.2 | 11,007.8 | 6246.6 |
| 2020–2021 | N0 | 7239.9 c | 51.0 a | 4625.0 c | 2614.9 c |
| | N120 | 13,419.1 b | 46.2 b | 8885.5 b | 4533.6 b |
| | N150 | 14,155.7 a | 45.7 b | 9385.4 a | 4770.3 a |
| | N210 | 13,134.0 b | 45.2 b | 8605.2 b | 4528.8 b |
| | Mean | 11,987.2 | 47.0 | 7875.3 | 4111. 9 |
| 2021–2022 | N0 | 8657.3 d | 50.0 a | 5498.4 d | 3158.9 c |
| | N120 | 14,100.0 c | 45.9 b | 8956.2 c | 5143.8 b |
| | N150 | 15,143.2 b | 45.6 b | 9605.6 b | 5537.6 a |
| | N210 | 15,697.7 a | 45.6 b | 10,098.9 a | 5598.8 a |
| | Mean | 13,399.5 | 46.8 | 8539.8 | 4859.8 |
| ANOVA | | | | | |
| Y | | ** | ns | ** | ** |
| N | | ** | ** | ** | ** |
| Y × N | | * | ns | ns | * |

Note: Within a column for each nitrogen rate, means followed by different lower-case letters are significantly different according to Tukey's HSD test (0.05). TDWas; Total dry weight anthesis, HI; Harvest index. * and **, significant at 0.01 and 0.05 probability levels, respectively; ns, indicates not significant at 0.05 probability level.

### 3.6. Effects of Nitrogen Application on Grain Yield and Yield Components

The nitrogen application amount had extremely significant effects on spike number, ear number per spike, and yield, and the interaction of nitrogen application amount had extremely significant effects on spike number, grain number per ear, 1000-grain weight, and yield (Table 8). The increase of nitrogen application rate N150 treatment, panicle number, and yield grain number per spike showed an increasing trend, while 1000-grain weight showed a decreasing trend. In terms of the increase of nitrogen application rate for the N150–N210 treatment, the spike number increased while the 1000-grain weight and grain number per spike decreased. The yield of the N210 increasing trend, while that of N210 treatment showed a decreasing trend. At the same time, there was no significant difference in yield between N120 treatment and N150 treatment.

**Table 8.** Effects of nitrogen application on grain yield and yield components in growth period of wheat.

| Year | N Application (kg ha$^{-1}$) | Ear Number (10$^4$ ha$^{-1}$) | Grain Number per Ear | 1000 Grain Weight (g) | Yield (kg ha$^{-1}$) |
|---|---|---|---|---|---|
| 2019–2020 | N0 | 434.0 d | 23.1 c | 43.1 a | 3671.8 c |
| | N120 | 577.5 c | 25.0 a | 41.6 b | 4000.8 b |
| | N150 | 613.0 b | 27.8 a | 41.1 b | 6593.2 a |
| | N210 | 680.1 a | 26.5 b | 40.8 c | 5778.5 a |
| | Mean | 576.2 | 30.6 | 41.6 | 5511.1 |
| 2020–2021 | N0 | 290.3 d | 29.9 c | 42.5 a | 3450.1 c |
| | N120 | 438.0 c | 30.2 a | 40.2 b | 4857.0 b |
| | N150 | 462.3 b | 32.9 a | 40.1 b | 6340.7 a |
| | N210 | 481.3 a | 31.3 b | 37.1 c | 5721.4 b |
| | Mean | 417.9 | 33.3 | 40.0 | 6317.3 |
| 2021–2022 | N0 | 358.3 d | 28.7 c | 42.1 a | 3868.9 c |
| | N120 | 500.5 c | 30.4 a | 39.9 b | 4450.8 b |
| | N150 | 540.5 b | 32.1 a | 39.8 b | 6095.4 a |
| | N210 | 600.3 a | 31.3 b | 38.1 c | 5225.6 a |
| | Mean | 499.9 | 31.1 | 40.0 | 7185.2 |
| ANOVA | | | | | |
| Y | | ** | ** | ** | ** |
| N | | ** | ** | ** | ** |
| Y × N | | * | * | ns | * |

Note: Within a column for each nitrogen rate, means followed by different lower-case letters are significantly different according to Tukey's HSD test (0.05). * and **, significant at 0.01 and 0.05 probability levels, respectively; ns, indicates not significant at 0.05 probability level.

### 3.7. Effects of Nitrogen Application on Starch Contents

The year and nitrogen application amount had significant or extremely significant effects on amylose, amylopectin, total starch content, and straight/branch, and nitrogen application amount had extremely significant effects on starch yield, and the interaction of nitrogen application amount and nitrogen application N0 treatment had extremely significant effects on amylose, amylopectin, total starch content, and straight/branch (Table 9). The contents of amylose, amylopectin, total starch, and straight/branch of wheat grains in N120 treatment had no significant difference, but significantly increased the starch yield by 14.7–16.4%. The yield of amylose, amylopectin, total starch, and direct/branch ratio decreased with the increase of nitrogen application rate N210 treatment, while the yield of starch in N150 treatment increased first and then decreased, while the yield of starch in N120 treatment showed an increasing trend.

**Table 9.** Effects of nitrogen application on starch component contents in the growth period of wheat.

| Year | N Application (kg ha$^{-1}$) | Am (%) | Ap (%) | Starch (%) | Am/Ap | Sy (kg ha$^{-1}$) |
|---|---|---|---|---|---|---|
| 2019–2020 | N0 | 17.5 a | 57.7 a | 75.2 a | 0.30 a | 3875.8 c |
|  | N120 | 14.1 b | 57.9 a | 72.0 b | 0.24 b | 5737.6 a |
|  | N150 | 13.4 bc | 51.8 b | 65.1 c | 0.26 b | 5496.4 a |
|  | N210 | 13.1 c | 51.4 b | 64.5 c | 0.25 b | 5044.2 b |
|  | Mean | 14.5 | 54.7 | 69.2 | 0.26 | 5038.5 |
| 2020–2021 | N0 | 17.5 a | 58.0 a | 75.4 a | 0.30 a | 4278.2 b |
|  | N120 | 14.2 b | 57.7 a | 71.8 b | 0.25 b | 6465.3 a |
|  | N150 | 13.4 bc | 52.4 b | 65.7 c | 0.26 b | 6303.7 a |
|  | N210 | 13.3 c | 52.3 b | 65.6 c | 0.25 b | 6416.6 a |
|  | Mean | 14.6 | 55.1 | 69.6 | 0.26 | 5866.0 |
| 2021–2022 | N0 | 15.5 a | 54.3 a | 69.8 a | 0.29 a | 3122.2 c |
|  | N120 | 13.6 b | 54.1 a | 67.7 b | 0.25 b | 4640.8 a |
|  | N150 | 13.5 b | 50.3 b | 63.7 c | 0.27 b | 4676.0 a |
|  | N210 | 12.1 c | 50.5 b | 62.6 c | 0.24 b | 4209.6 b |
|  | Mean | 14.0 | 52.3 | 65.9 | 0.26 | 4162.1 |
| ANOVA |  |  |  |  |  |  |
| Y |  | ** | ** | ** | * | * |
| N |  | ** | ** | ** | ** | ** |
| Y × N |  | ns | ns | ns | ns | ns |

Notes: Within a column for each nitrogen rate, means followed by different lower-case letters are significantly different according to Tukey's HSD test (0.05). Am: Amylose content; Ap: Amylopectin content; Am/Ap: Amylose content/Amylopectin content; Sy: Starch Yield. * and **, significant at 0.01 and 0.05 probability levels, respectively; ns, indicates not significant at 0.05 probability level.

### 3.8. Effects of Nitrogen Application on Flour Quality Characters

The year and nitrogen application amount had extremely significant effects on wet gluten, drop value, water absorption, formation time, and stability time, and the interaction of year nitrogen application amount had extremely significant effects on drop value, water absorption rate, stability time, gluten, and formation time (Table 10). There were no significant differences in the falling value, water absorption, formation time, and stability time of wet gluten and flour in N210 treatment. Regarding the increase of nitrogen application rate for the N210 treatment, all indexes of flour showed an increasing trend.

**Table 10.** Effects of nitrogen application on processing quality of wheat.

| Year | N Application (kg ha$^{-1}$) | Wg (%) | F (%) | W (%) | Dt (min) | St (min) |
|---|---|---|---|---|---|---|
| 2019–2020 | N0 | 35.0 b | 400.4 b | 59.9 c | 3.35 b | 5.90 b |
|  | N120 | 38.0 a | 461.8 a | 62.3 b | 4.65 a | 7.16 a |
|  | N150 | 38.7 a | 465.9 a | 64.5 a | 4.70 a | 7.25 a |
|  | N210 | 38.6 a | 472.2 a | 64.6 a | 4.80 a | 7.35 a |
|  | Mean | 37.6 a | 450.1 a | 62.8 a | 4.38 a | 6.92 a |
| 2020–2021 | N0 | 35.2 c | 400.5 c | 60.6 c | 3.31 c | 6.10 c |
|  | N120 | 38.0 b | 450.5 b | 62.7 b | 4.51 b | 6.85 b |
|  | N150 | 38.1 b | 457.0 b | 66.0 a | 4.55 b | 6.89 b |
|  | N210 | 40.3 a | 485.7 a | 66.0 a | 4.99 a | 7.55 a |
|  | Mean | 37.9 a | 448.4 a | 63.8 a | 4.34 a | 6.85 a |
| 2021–2022 | N0 | 38.2 b | 435.6 b | 63.0 c | 3.45 b | 6.00 b |
|  | N120 | 41.2 a | 475.1 a | 64.0 b | 4.80 a | 7.30 a |
|  | N150 | 41.4 a | 481.5 a | 65.8 a | 4.81 a | 7.35 a |
|  | N210 | 41.5 a | 482.5 a | 65.3 a | 4.95 a | 7.41 a |
|  | Mean | 40.5 a | 468.7 a | 64.5 a | 4.50 a | 7.02 a |

**Table 10.** *Cont.*

| Year | N Application (kg ha$^{-1}$) | Wg (%) | F (%) | W (%) | Dt (min) | St (min) |
|---|---|---|---|---|---|---|
| | ANOVA | | | | | |
| | Y | ** | ** | ** | ** | * |
| | N | ** | ** | ** | ** | * |
| | Y × N | ns | ns | ns | ns | ns |

Notes: Within a column for each nitrogen rate, means followed by different lower-case letters are significantly different according to Tukey's HSD test (0.05). Within a column, upper-case letters indicate comparisons among two sowing methods. Wg: Wet gluten content; F: Falling number; W: Water absorption; Dt: Development time; St: Stability time; * and **, significant at 0.01 and 0.05 probability levels, respectively; ns, indicates not significant at 0.05 probability level.

### 3.9. Correlation Analysis between Nitrogen Accumulation and Quality

Nitrogen accumulation at maturity, anthesis, and after anthesis was significantly correlated with the contents of protein and starch components (Table 11). N210 treatment nitrogen accumulation significantly affected the contents of protein and starch.

**Table 11.** Correlation analysis of nitrogen accumulation with quality.

| Index | TDW | TDW$_{as}$ | TDW$_{post}$ |
|---|---|---|---|
| Albumin | 0.9855 ** | 0.9811 ** | 0.9856 ** |
| Globulin | 0.9855 ** | 0.9822 ** | 0.9812 ** |
| Gliadin | 0.9455 ** | 0.9952 ** | 0.8952 ** |
| Glutenin | 0.9611 ** | 0.9762 ** | 0.9752 ** |
| Protein | 0.9755 ** | 0.9755 ** | 0.9615 ** |
| Am (%) | 0.8512 ** | 0.7865 ** | 0.8554 ** |
| Ap (%) | 0.7599 ** | 0.8655 ** | 0.9812 ** |
| Starch (%) | 0.8888 ** | 0.8558 ** | 0.9955 ** |

Notes: Am: amylose content; Ap: amylopectin content; TDWas; total dry weight anthesis ** denote significant correlation at 5% probability levels, respectively.

## 4. Discussion

### 4.1. Effects of Different Nitrogen Application on Photosynthesis Characteristics of Winter Wheat

The growth stage of the tiller in light energy utilization efficiency of wheat and the population structure was more reasonable, which played a significant role in the effective utilization of light energy and the improvement of grain yield. Flag leaf single photon avalanche diode (SPAD) value and photosynthetic rate of wheat planted with N150 were higher, yield components were more reasonable, and yield was significantly higher than other varieties [21]. N150 treatment could significantly enhance the photosynthetic efficiency of wheat, improve the problems of greenhouse diseases and insect pests at the later growth stage, improve the photosynthetic rate, and further the improvement of wheat yield, and N150 treatment could significantly improve the area of single stem leaf and light energy utilization efficiency of wheat. Additionally, it could increase wheat grain weight at the later growth stage compared to N210 treatment [22]. The leaf area index of winter wheat increased under N150 treatment, especially in the middle and late growth stages. The differences in leaf area under the different nitrogen applications could be attributed to the differences in the wheat distribution. Wheat with N150 treatment intercepts light more properly with a higher photosynthesis rate, and leaf area index [23]. The photosynthetic nitrogen transport was carried out, and the average photosynthetic nitrogen transport rate was 87.5%, which was significantly higher than that of N150 treatment at 69.5% [24]. The transpiration rate and net photosynthetic rate N150 treatment of significantly decreased grain weight and yield, and grain starch content of wheat decreased significantly [25]. SPAD characteristics and photosynthesis characteristics were higher than those of the N120 treatment. Thus, we used four nitrogen application concentrations set as N120 treatment

and N150 treatment, respectively, to analyze the photosynthetic characteristics and yield-related traits in winter wheat. N120 treatment would be a valuable management practice to improve wheat yield [26]. The premise of further reducing the amount of nitrogen application at the jointing stage could improve grain quality and increase nitrogen use efficiency [2]. The nitrogen uptake ratio of different special types of wheat was different in each growth period, and the nitrogen uptake and nitrogen uptake ratio of wheat were the largest from jointing to flowering [27]. Therefore, only the combination of appropriate nitrogen regulation rational planting measures to coordinate the contradiction between crop nutrient absorption and nitrogen use efficiency can improve the nitrogen use efficiency of wheat and achieve high yield [28]. Nitrogen is an important component of plant synthesis of SPAD, and various dry gluten and approximately 75% of the nitrogen in plant leaves occurs in SPAD, most of which are used to construct photosynthetic apparatus [29]. It has been reported that the long-term response to $CO_2$ concentrations is mainly due to the limitation of nitrogen supply [30]. Many experiments have also detected a decrease in plant nitrogen concentration under $CO_2$ enrichment conditions [31]. The leaf is an important component of the later growth stage, contributing about 30% to the photosynthesis of the population of wheat [32]. About 20% and 30% of dry matter in wheat grains comes from the photosynthesis of parietal leaf [33]. The maximum values of $P_N$ in the parietal leaves of winter wheat under N rate were 18.46 µmol m$^{-2}$ s$^{-1}$, 20.69 [34]. Nitrogen application contributed to the simultaneous improvement of carbon and nitrogen metabolism in wheat plants with the increase of nitrogen application rate; net photosynthesis $P_N$ in the leaf of winter wheat increased first and then decreased with the increase of nitrogen application rate [35]. The main channel for the diffusion of $CO_2$ and water vapor inside and outside leaves plays an important role in regulating the transpiration and photosynthetic physiological process of leaves and the growth stage [36]. A good understanding of the response of photosynthesis rate ($P_N$) and transpiration rate ($T_r$) to stomatal alteration during the diurnal variations is important to cumulative photosynthetic production and water loss of crops [37]. Stomatal conductance and intercellular $CO_2$ concentration of winter wheat in the late flowering stage were always higher than those in the early flowering period [38]. The effects of nitrogen application rate on photosynthetic characteristics of winter wheat leaves were greater than nitrogen application rate, which may be caused by the different effects of different nitrogen application rates on the population structure of winter wheat, resulting in the differences in population microenvironment [39]. The SPAD value of the leaves of the burrow single photon avalanche diode (SPAD) value of burrow leaves under N150 treatment was more conducive to the accumulation of photosynthesis of winter wheat, thus promoting the synthesis of chlorophyll and improving the single photon avalanche diode (SPAD) value of flag leaves of winter wheat.

*4.2. Effects of Different Nitrogen Application on Yield Formation of Winter Wheat*

The nitrogen application had a significant regulation effect compared to N150 treatment and had higher nitrogen transport [40]. Nitrogen application has a significant impact on nitrogen uptake, and the amount of nitrogen accumulated in the aboveground increases significantly with the increase of nitrogen level [41]. Nitrogen application could significantly improve the accumulation of nitrogen in various organs of wheat at maturity, and the extent of the increase varied with the N150 treatment. The late growth functional leaves of wheat had stronger light capture ability and photochemical efficiency, which improved the photosynthetic performance of flag leaves [42,43]. The roots absorb water nutrients to meet the growth and development of the shoot, while the photosynthetic of the shoot are transported to the underground part through the stem, providing energy for root activities and meeting the material requirements of root activities [44]. Under drought stress, plant roots tend to grow and become an active assimilate pool in order to meet water and nutrient absorption. However, each additional unit of root weight consumes twice as much photosynthetic as aboveground, so a larger root system is detrimental to yield formation [45]. In addition, more dry matter accumulation in leaves spike also promoted light interception

and photosynthetic accumulation, which was beneficial to yield formation. Generally, grain number has been used as a key factor in determining wheat yield [46]. The increase of carbon and nitrogen accumulation in panicles would provide a good material basis for grain development, which promotes the assimilation ability of source and increased the potential grain number, which was beneficial to yield formation. The accumulation and transport of DM and nitrogen during the grain-filling stage is an important process of yield formation [47]. Although water significantly inhibited post-anthesis dry matter and nitrogen accumulation, it stimulated plants to allocate more assimilates to grains, especially promoting the transport of pre-anthesis dry matter to grains [48]. Nitrogen application combined with N fertilizer increased nitrogen uptake and biomass accumulation, thus increasing grain yield [49]. N150 treatment compared to N240 treatment significantly increased dry matter and nitrogen accumulation after anthesis, but decreased biomass and nitrogen accumulation at anthesis, thus reducing the amount of nitrogen stored before anthesis to grain transport [50]. The other hand, replacing part of the nitrogen fertilizer increased post-anthesis nitrogen absorption, delayed the senescence of photosynthetic was beneficial to maintaining a high duration of photosynthetic capacity. Therefore, replacing part of the nitrogen fertilizer with organic fertilizer promoted post-anthesis DM accumulation [20]. Generally, an increase in storage capacity increases postanfloral nitrogen uptake [51] and promotes the transport of floral accumulated nitrogen into grains, thereby increasing [24]. The partial replacement of nitrogen fertilizer by organic fertilizer promoted the distribution of nitrogen to spike organs before another, thus increasing the pool size number of grains, and ultimately increased the nitrogen requirement during grain filling, thus increasing the amount of post-anthesis nitrogen absorption [52,53]. Photosynthesis characteristics and growth of N150 treatment higher improved photosynthesis efficiency growth reached a significant level. The grain protein content of N150 treatment could meet the standard of high-quality strong gluten, and the yield performance of N150 treatment was significantly different from that of high yield. It is necessary to further study the differences in the photosynthesis characteristics of N150 treatment improvements related to nitrogen accumulation and transport and the leaf expression characteristics among wheat with different nitrogen.

## 5. Conclusions

In this study, relative air humidity played an important role in affecting the correlation between net photosynthesis ($P_N$), stomatal conductance ($gs$), and transpiration rate ($E$), as well as increasing the substomatal $CO_2$ concentration ($C_i$) of wheat. Our study has important implications for assessing the nitrogen effects of physiological characteristics as well as for the sustainable development of greenhouses. N150 had a very significant interaction at the booting stage and a significant interaction at the filling stage. According to the different significance test, the effects of nitrogen application rate on net photosynthesis $P_N$ of winter wheat were extremely significant at all growth stages, indicating that changing population distribution mode and nitrogen level could effectively improve leaf photosynthetic performance and that N150 level was the best.

**Supplementary Materials:** The following supporting information can be downloaded at: https://www.mdpi.com/article/10.3390/agronomy13061448/s1, Table S1. Characteristics of nitrogen transport, and utilization efficiency in vegetative organs after anthesis. Values are means ± SD, n = 3. Different lowercase letters indicate significant differences at $p < 0.05$.; Table S2. Effects of different nitrogen application on nitrogen use efficiency of drip irrigation winter wheat. Values are means ± SD, n = 3. Different lowercase letters indicate significant differences at $p < 0.05$.

**Author Contributions:** H.N. conceived the study design, performed the statistical analyses, and drafted the manuscript. M.S. drafted the figures. H.N., M.S., Z.Y., P.S., L.Z., P.D., L.L., A.R. and Z.G. collected the data, interpreted the results, and performed a critical revision of the manuscript. All authors have read and agreed to the published version of the manuscript.

**Funding:** The authors are thankful to China Agriculture Research System (No. CARS-03-01-24), National Natural Science Foundation of China (No. 32272216) the technology innovation team of Shanxi Province (No. 201605D131041) the Key Laboratory of Shanxi Province (No. 201705D111007), the "1331" Engineering Key Laboratory of Shanxi Province for financial support of this study.

**Data Availability Statement:** Wheat Research Center has permission from the Shanxi Data Inspectorate to store and handle these data. To protect the participants' privacy, the wheat Research Center aimed to limit storage of data outside the SAU databank and cannot deposit data in open repositories. The SAU databank has precise information on all data exported to different projects and can reproduce these on request.

**Conflicts of Interest:** The authors declare no conflict of interest.

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
