# Peer review of "Effects of Nitrogen on Photosynthetic Productivity and Yield Quality of Wheat (Triticum aestivum L.)"

_agronomy, doi:10.3390/agronomy13061448_

Round 1

Reviewer 1 Report (New Reviewer)

1. General Comments

 The paper is interesting.  The work presents a very deeply review about vegetal physiological concerning productivity. The authors applied this knowledge to understand the effects of nitrogen on photosynthetic productivity and yield quality in an important crop, Triticum aestivum L. So, as a scientific reader, I find the paper very appealing, motivating and understandable.

2. Section by section

2.1. Introduction:

The introduction is very easy to read, very comprehensible and has a lot of references to consolidate the affirmations made.

2.2. Material and Methods:

Material and Methods are, from my point view, well conducted. The experimental design and  statistical methods used are adequate In case of interest/need, the description of methods used allows to replicate the assay.

2.3. Results:

Results are well presented and the presentation is easy to understand.

2.4. Discussion:

Discussion is well conducted and interesting to read, it is well supported in bibliographic references and clearly explains the observed results.

3. Other comments

I appreciated the article; it was very pleasant to read. It is a source of reliable information, with practical application.

Author Response

Agronomy-2388545

Effects of Nitrogen on Photosynthetic Productivity and Yield Quality of Wheat (Triticum aestivum L.)

Dear Reviewer

Thank you very much for handling our manuscript. Thank you for your suggestion. Thank you for your decision and constructive comments on our manuscript. We had considered your suggestion carefully and made some changes. In the revised manuscript, we have attempted to address all the comments from the reviewers. All the changes that we have made in the revised manuscript are highlighted with track Changes in the track file. Below are our detailed point-to-point responses to the comments from the reviewers.

Thank you for your kind consideration.

Best regards,

Dr. Hafeez Noor

Key Laboratory of Functional Agriculture in the Loess Plateau, Ministry of Agriculture and Rural Affairs, Taigu,

College of Agronomy, Shanxi Agricultural University,

Taigu, Shanxi, 030801, P. R China

Email: hafeeznoorbaloch@gmail.com

*********************************************************************

Review Report 1

  1. General Comments

 The paper is interesting.  The work presents a very deeply review about vegetal physiological concerning productivity. The authors applied this knowledge to understand the effects of nitrogen on photosynthetic productivity and yield quality in an important crop, Triticum aestivum L. So, as a scientific reader, I find the paper very appealing, motivating and understandable.

 Response: Thank you for your suggestion. The manuscript has benefited from these insightful suggestions. Thank you for your decision and constructive comments on our manuscript. We had considered your suggestion carefully and made some changes. In the revised manuscript, we have attempted to address all the comments from the reviewers. Below are our detailed point-to-point responses to the comments from the reviewers.

  1. Section by section

2.1. Introduction:

The introduction is very easy to read, very comprehensible and has a lot of references to consolidate the affirmations made.

Response: Thank you for your suggestion. The manuscript has benefited from these insightful suggestions.

2.2. Material and Methods:

Material and Methods are, from my point view, well conducted. The experimental design and statistical methods used are adequate In case of interest/need, the description of methods used allows to replicate the assay.

 Response: Thank you for your suggestion. We had considered your suggestion carefully and made some changes.

2.3. Results:

Results are well presented and the presentation is easy to understand.

 Response: Thank you for your suggestion. The manuscript has benefited from these insightful suggestions. Thank you for your decision and constructive comments on our manuscript. We had considered your suggestion carefully and made some changes. In the revised manuscript, we have attempted to address all the comments from the reviewers.

2.4. Discussion:

Discussion is well conducted and interesting to read, it is well supported in bibliographic references and clearly explains the observed results.

 Response: Thank you for your suggestion. The manuscript has benefited from these insightful suggestions. Thank you for your decision and constructive comments on our manuscript.

  1. Other comments

I appreciated the article; it was very pleasant to read. It is a source of reliable information, with practical application.

 Response: Thank you for your suggestion. The manuscript has benefited from these insightful suggestions. In the revised manuscript, we have attempted to address all the comments from the reviewers. Below are our detailed point-to-point responses to the comments from the reviewers.

Reviewer 2 Report (New Reviewer)

Dear authors,

I believe that the experimental design is rigorous and the investigation that you are reporting is scientifically sound. Please, consider the following comments to improve the readability of the manuscript:

-          Line 12: Please revise the grammar of the first sentence of the abstract: “This study means effects…”. I suggest to start as “This study assesses the effects of nitrogen…”. Additionally, strongly recommend to included a brief description of the background of the present work. Please, include 2 lines of background of the present work (which is the problem that you are addressing? Why is it urgent? How other have attempted to deal with it?) at the very beginning of the abstract.

-          Line 15 to 18: Please, revise the connection between these 2 sentences. Currently there is a stop in the middle of the 2 sentences that should be better linked.

-          Line 19: Missing brackets for the definition of single photon avalanche diode (SPAD).

-          Line 18 to 21: The sentence needs to be revised as it is very hard to read. Additionally, please avoid including these detailed descriptions of the results in the abstract and instead devote more work to the background of your investigation.

-          Line 21 to 23: The last sentence should be at the beginning of the abstract. Additionally, I still suggest the authors to include a brief explanation of the background of their work.

-          Line 24: Please, avoid the use of abbreviations as keyword, unless they have been strongly stablished in your field.

-          Lines 66 to 72: Although the readability of the introduction has improved compared to the abstract, it still can be polished. Avoid to use “however” too many times and instead employ “nevertheless”, “but”, etc.

-          Lines 76 to 85: It is good that you have included a brief paragraph at the end of the introduction to focus the article on the matter that you are going to investigate in the subsequent sections. I encourage to revise the firsts sentences of this paragraph because they look more about a general information on the background of the project that should not be included in the last paragraph of the introduction. Please, move this information early in the introduction and use this paragraph to clarify the problem that you are addressing, the approach that you are following, and why is it novel.

-          Line 96: superscript missing in m2

-          Line 99: missing space in 20 cm and superscript in 99th

-          Line 136 to 138: You are describing the use of Origin lab for the preparation of a Figure but the only figure available in the manuscript is a photograph, not a plot that can be prepared in Origin.

-          Line 122: SPAD needs to be defined in the first appearance in the text (in addition to the abstract).

-          Lines 214 and 215: You are defining here again SPAD. This should be defined only once earlier in the text, in addition to the abstract. Please apply this logic to all the other abbreviations.

-          Lines 282 and 283: Missing the definitive article at the beginning of the sentence: The N150 treatment…

-          Line 284: Remove capital letter in dry matter or it would be better to define the abbreviation (DM) and use it consistently throughout the whole manuscript.

-          Line 300: The header of 1 of the columns of table 8 is “Thousand grain weight” but this parameters is mentioned in the text as 1000-grain weight. Please, unify the nomenclature. Also, provide an explanation of the information that this parameter provide, as this will increase the accessibility of the manuscript to non-expert-readers.

-          Line 340: Please, define the abbreviation TDW earlier in the text. The readers would appreciate some of these tables as plots for easier interpretation.

-          Line 452 to 456: in the discussion, about repetition and redundances in the sentences. Please revise and reorganize these sentences to avoid the excessive use of although.

-          Line 482 to 495: The conclusions paragraph is acceptable although it contains some discussive elements and establishment of relations between parameter that could be better placed in the previous discussion section. Additionally, the last sentence of this paragraph is more suitable for the beginning of the paragraph, as it would act as reminder of the novelty and importance of the present study before listing the key conclusions of the investigation.

Dear authors,

I have provided several comments to improve the readability. Particularly, the abstract require major improvements:

-          Line 12: Please revise the grammar of the first sentence of the abstract: “This study means effects…”. I suggest to start as “This study assesses the effects of nitrogen…”. Additionally, strongly recommend to included a brief description of the background of the present work. Please, include 2 lines of background of the present work (which is the problem that you are addressing? Why is it urgent? How other have attempted to deal with it?) at the very beginning of the abstract.

-          Line 15 to 18: Please, revise the connection between these 2 sentences. Currently there is a stop in the middle of the 2 sentences that should be better linked.

-          Line 19: Missing brackets for the definition of single photon avalanche diode (SPAD).

-          Line 18 to 21: The sentence needs to be revised as it is very hard to read. Additionally, please avoid including these detailed descriptions of the results in the abstract and instead devote more work to the background of your investigation.

-          Line 21 to 23: The last sentence should be at the beginning of the abstract. Additionally, I still suggest the authors to include a brief explanation of the background of their work.

Author Response

Agronomy-2388545

Effects of Nitrogen on Photosynthetic Productivity and Yield Quality of Wheat (Triticum aestivum L.)

Dear Reviewer

Thank you very much for handling our manuscript. Thank you for your suggestion. Thank you for your decision and constructive comments on our manuscript. We had considered your suggestion carefully and made some changes. In the revised manuscript, we have attempted to address all the comments from the reviewers. All the changes that we have made in the revised manuscript are highlighted with track Changes in the track file. Below are our detailed point-to-point responses to the comments from the reviewers.

Thank you for your kind consideration.

Best regards,

Dr. Hafeez Noor

Key Laboratory of Functional Agriculture in the Loess Plateau, Ministry of Agriculture and Rural Affairs, Taigu,

College of Agronomy, Shanxi Agricultural University,

Taigu, Shanxi, 030801, P. R China

Email: hafeeznoorbaloch@gmail.com

*********************************************************************

Review Report-2

Dear authors,

I believe that the experimental design is rigorous and the investigation that you are reporting is scientifically sound. Please, consider the following comments to improve the readability of the manuscript:

 Line 12: Please revise the grammar of the first sentence of the abstract: “This study means effects…”. I suggest to start as “This study assesses the effects of nitrogen…”. Additionally, strongly recommend to included a brief description of the background of the present work. Please, include 2 lines of background of the present work (which is the problem that you are addressing? Why is it urgent? How other have attempted to deal with it?) at the very beginning of the abstract.

Response: Thank you for decision and constructive comments on our manuscript. We had considered your suggestion carefully and made some changes Line 12

Line 15 to 18: Please, revise the connection between these 2 sentences. Currently there is a stop in the middle of the 2 sentences that should be better linked.

Response: Thank you for decision and constructive comments on our manuscript. We had considered your suggestion carefully and made some changes Line 15 to 18:

Line 19: Missing brackets for the definition of single photon avalanche diode (SPAD).

Response: Thank you for decision and constructive comments on our manuscript. We had considered your suggestion carefully and made some changes Line 19:

Line 18 to 21: The sentence needs to be revised as it is very hard to read. Additionally, please avoid including these detailed descriptions of the results in the abstract and instead devote more work to the background of your investigation.

Response: Thank you for decision and constructive comments on our manuscript. We had considered your suggestion carefully and made some changes Line 18 to 21:

 Line 21 to 23: The last sentence should be at the beginning of the abstract. Additionally, I still suggest the authors to include a brief explanation of the background of their work.

Response: Thank you for decision and constructive comments on our manuscript. We had considered your suggestion carefully and made some changes Line 21 to 23:

Line 24: Please, avoid the use of abbreviations as keyword, unless they have been strongly stablished in your field.

Response: Thank you for decision and constructive comments on our manuscript. We had considered your suggestion carefully and made some changes Line 24:

Lines 66 to 72: Although the readability of the introduction has improved compared to the abstract, it still can be polished. Avoid to use “however” too many times and instead employ “nevertheless”, “but”, etc.

Response: Thank you for decision and constructive comments on our manuscript. We had considered your suggestion carefully and made some changes Lines 66 to 72:

Lines 76 to 85: It is good that you have included a brief paragraph at the end of the introduction to focus the article on the matter that you are going to investigate in the subsequent sections. I encourage to revise the firsts sentences of this paragraph because they look more about a general information on the background of the project that should not be included in the last paragraph of the introduction. Please, move this information early in the introduction and use this paragraph to clarify the problem that you are addressing, the approach that you are following, and why is it novel.

Response: Thank you for decision and constructive comments on our manuscript. We had considered your suggestion carefully and made some changes Lines 76 to 85:

Line 96: superscript missing in m2

Response: Thank you for decision and constructive comments on our manuscript. We had considered your suggestion carefully and made some changes Line 96

Line 99: missing space in 20 cm and superscript in 99th

Response: Thank you for decision and constructive comments on our manuscript. We had considered your suggestion carefully and made some changes Line 99

Line 136 to 138: You are describing the use of Origin lab for the preparation of a Figure but the only figure available in the manuscript is a photograph, not a plot that can be prepared in Origin.

Response: Thank you for decision and constructive comments on our manuscript. We had considered your suggestion carefully and made some changes Line 136 to 138:

Line 122: SPAD needs to be defined in the first appearance in the text (in addition to the abstract).

Response: Thank you for decision and constructive comments on our manuscript. We had considered your suggestion carefully and made some changes Line 122

Lines 214 and 215: You are defining here again SPAD. This should be defined only once earlier in the text, in addition to the abstract. Please apply this logic to all the other abbreviations.

Response: Thank you for decision and constructive comments on our manuscript. We had considered your suggestion carefully and made some changes Lines 214 and 215:

 Lines 282 and 283: Missing the definitive article at the beginning of the sentence: The N150 treatment…

Response: Thank you for decision and constructive comments on our manuscript. We had considered your suggestion carefully and made some changes Lines 282 and 283:

 Line 284: Remove capital letter in dry matter or it would be better to define the abbreviation (DM) and use it consistently throughout the whole manuscript.

Response: Thank you for decision and constructive comments on our manuscript. We had considered your suggestion carefully and made some changes Line 284

Line 300: The header of 1 of the columns of table 8 is “Thousand grain weight” but this parameters is mentioned in the text as 1000-grain weight. Please, unify the nomenclature. Also, provide an explanation of the information that this parameter provide, as this will increase the accessibility of the manuscript to non-expert-readers.

Response: Thank you for decision and constructive comments on our manuscript. We had considered your suggestion carefully and made some changes Line 300:

 Line 340: Please, define the abbreviation TDW earlier in the text. The readers would appreciate some of these tables as plots for easier interpretation.

Response: Thank you for decision and constructive comments on our manuscript. We had considered your suggestion carefully and made some changes Line 340:

 Line 452 to 456: in the discussion, about repetition and redundances in the sentences. Please revise and reorganize these sentences to avoid the excessive use of although.

Response: Thank you for decision and constructive comments on our manuscript. We had considered your suggestion carefully and made some changes Line 452 to 456:

 Line 482 to 495: The conclusions paragraph is acceptable although it contains some discussive elements and establishment of relations between parameter that could be better placed in the previous discussion section. Additionally, the last sentence of this paragraph is more suitable for the beginning of the paragraph, as it would act as reminder of the novelty and importance of the present study before listing the key conclusions of the investigation.

Response: Thank you for your suggestion. Thank you for decision and constructive comments on our manuscript. We had considered your suggestion carefully and made some changes. We have tried our best to improve and made some marks in the manuscript.

Reviewer 3 Report (New Reviewer)

- Abstract

I suggest that the summary be enlarged. Explore the results further.

Keywords cannot be repeated in the title. To review.

-Introduction

There are many important information that are in need of references. Insert references.

A hypothesis must be formulated for the study. Information before the objectives is obvious, I suggest withdrawing.

Objectives have redundant information. I suggest redoing.

- Material and methods

Insert references or equipment used to collect climate data in the region.

Mention the source of N used and the percentage of N present in this source.

There is no need to mention which software was used to produce the Figure. Withdraw that information.

Why polynomial regression analysis was not used for quantitative levels (doses). Tukey's test in this case would not be applicable, as a difference in responses is already expected.

- Results

Replace all tables with figures with polynomial regression analysis. Measure the production efficiency of each variable, using the equation built with the regression model.

For the variables that had interaction between the factors, a three-dimensional figure is indicated, with the x, y and z axes.

Author Response

Agronomy-2388545

Effects of Nitrogen on Photosynthetic Productivity and Yield Quality of Wheat (Triticum aestivum L.)

Dear Reviewer

Thank you very much for handling our manuscript. Thank you for your suggestion. Thank you for your decision and constructive comments on our manuscript. We had considered your suggestion carefully and made some changes. In the revised manuscript, we have attempted to address all the comments from the reviewers. All the changes that we have made in the revised manuscript are highlighted with track Changes in the track file. Below are our detailed point-to-point responses to the comments from the reviewers.

Thank you for your kind consideration.

Best regards,

Dr. Hafeez Noor

Key Laboratory of Functional Agriculture in the Loess Plateau, Ministry of Agriculture and Rural Affairs, Taigu,

College of Agronomy, Shanxi Agricultural University,

Taigu, Shanxi, 030801, P. R China

Email: hafeeznoorbaloch@gmail.com

*********************************************************************

Review Report-3

- Abstract

I suggest that the summary be enlarged. Explore the results further.

Keywords cannot be repeated in the title. To review.

 Response: Thank you for your suggestion. The manuscript has benefited from these insightful suggestions. Thank you for your decision and constructive comments on our manuscript. We had considered your suggestion carefully and made some changes.

-Introduction

There are many important information that are in need of references. Insert references.

A hypothesis must be formulated for the study. Information before the objectives is obvious, I suggest withdrawing.

Objectives have redundant information. I suggest redoing.

Response: Thank you for your suggestion. The manuscript has benefited from these insightful suggestions. Thank you for your decision and constructive comments on our manuscript. We had considered your suggestion carefully and made some changes. In the revised manuscript, we have attempted to address all the comments from the reviewers. Below are our detailed point-to-point responses to the comments from the reviewers.

- Material and methods

Insert references or equipment used to collect climate data in the region.

Mention the source of N used and the percentage of N present in this source.

There is no need to mention which software was used to produce the Figure. Withdraw that information.

Why polynomial regression analysis was not used for quantitative levels (doses). Tukey's test in this case would not be applicable, as a difference in responses is already expected.

Response: Thank you for your decision and constructive comments on our manuscript. We had considered your suggestion carefully and made some changes. In the revised manuscript, we have attempted to address all the comments from the reviewers.

- Results

Replace all tables with figures with polynomial regression analysis. Measure the production efficiency of each variable, using the equation built with the regression model.

For the variables that had interaction between the factors, a three-dimensional figure is indicated, with the x, y and z axes.

Response: Thank you for your suggestion. The manuscript has benefited from these insightful suggestions. Thank you for your decision and constructive comments on our manuscript. We had considered your suggestion carefully and made some changes. In the revised manuscript, we have attempted to address all the comments from the reviewers. Below are our detailed point-to-point responses to the comments from the reviewers.

Round 2

Reviewer 2 Report (New Reviewer)

Dear authors,

Thank you for your work. I think you have addressed correctly some of my comments but you have ignored or misunderstood others. For instance, you have not included a brief description of the background of your investigation in the abstract. If you decide to address this comment, please do not increase further the length of the abstract and rather delete some of the descriptions of the results in order to make space for the brief background at the beginning of the abstract. Also, I do not see the reason for including “dry matter” among the keywords. I suggest to use “however” in line 70 and “nevertheless” in line 74. Please define the abbreviation of single-photon avalanche diode (SPAD) again in the main manuscript that follows the abstract. This is literally what is written in the instructions for authors of the journal: Acronyms/Abbreviations/Initialisms should be defined the first time they appear in each of three sections: the abstract; the main text; the first figure or table. When defined for the first time, the acronym/abbreviation/initialism should be added in parentheses after the written-out form. Additionally, I suggested to include some figures rather than all data. Although this matter could be improved, I believe that the revised version is much better than the previous one.

Best wishes

I suggest to use “however” in line 70 and “nevertheless” in line 74

Author Response

Agronomy-2388545

Effects of Nitrogen on Photosynthetic Productivity and Yield Quality of Wheat (Triticum aestivum L.)

Dear Reviewer

Thank you very much for handling our manuscript. Thank you for your suggestion. Thank you for your decision and constructive comments on our manuscript. We had considered your suggestion carefully and made some changes. In the revised manuscript, we have attempted to address all the comments from the reviewers. All the changes that we have made in the revised manuscript. Below are our detailed point-to-point responses to the comments from the reviewers.

Thank you for your kind consideration.

Best regards,

Dr. Hafeez Noor

Key Laboratory of Functional Agriculture in the Loess Plateau, Ministry of Agriculture and Rural Affairs, Taigu,

College of Agronomy, Shanxi Agricultural University,

Taigu, Shanxi, 030801, P. R China

Email: hafeeznoorbaloch@gmail.com

*********************************************************************

Review Report-2

Thank you for your work. I think you have addressed correctly some of my comments but you have ignored or misunderstood others. For instance, you have not included a brief description of the background of your investigation in the abstract. If you decide to address this comment, please do not increase further the length of the abstract and rather delete some of the descriptions of the results in order to make space for the brief background at the beginning of the abstract. Also, I do not see the reason for including “dry matter” among the keywords. I suggest to use “however” in line 70 and “nevertheless” in line 74. Please define the abbreviation of single-photon avalanche diode (SPAD) again in the main manuscript that follows the abstract. This is literally what is written in the instructions for authors of the journal: Acronyms/Abbreviations/Initialisms should be defined the first time they appear in each of three sections: the abstract; the main text; the first figure or table. When defined for the first time, the acronym/abbreviation/initialism should be added in parentheses after the written-out form. Additionally, I suggested to include some figures rather than all data. Although this matter could be improved, I believe that the revised version is much better than the previous one.

Best wishes

I suggest using “however” in line 70 and “nevertheless” in line 74

Response: Thank you for your suggestion. Thank you for your decision and constructive comments on our manuscript. We had considered your suggestion carefully and made some changes. We have tried our best to improve and made some marks in the manuscript.

Reviewer 3 Report (New Reviewer)

The authors made most of the proposed suggestions.

Author Response

Agronomy-2388545

Effects of Nitrogen on Photosynthetic Productivity and Yield Quality of Wheat (Triticum aestivum L.)

Dear Reviewer

Thank you very much for handling our manuscript. Thank you for your suggestion. Thank you for your decision and constructive comments on our manuscript. We had considered your suggestion carefully and made some changes. In the revised manuscript, we have attempted to address all the comments from the reviewers. All the changes that we have made in the revised manuscript file. Below are our detailed point-to-point responses to the comments from the reviewers.

Thank you for your kind consideration.

Best regards,

Dr. Hafeez Noor

Key Laboratory of Functional Agriculture in the Loess Plateau, Ministry of Agriculture and Rural Affairs, Taigu,

College of Agronomy, Shanxi Agricultural University,

Taigu, Shanxi, 030801, P. R China

Email: hafeeznoorbaloch@gmail.com

*********************************************************************

Response: Thank you for your suggestion. The manuscript has benefited from these insightful suggestions. Thank you for your decision and constructive comments on our manuscript. We had considered your suggestion carefully and made some changes. In the revised manuscript, we have attempted to address all the comments from the reviewers. Below are our detailed point-to-point responses to the comments from the reviewers.

This manuscript is a resubmission of an earlier submission. The following is a list of the peer review reports and author responses from that submission.

Round 1

Reviewer 1 Report

Review Report: Long Term Effects of Nitrogen Fertilizer on Photosynthetic Productivity of Yield Quality Wheat (Triticum aestivum L.) in Response to Nitrogen Application Levels

The paper examines the effect of different nitrogen levels on photosynthetic productivity and yield quality of wheat. There is English language and editing issues which needs to be addressed before the paper can be reviewed.  

1.     The title needs to be shortened to make it clearer. For example, something like “Effect of Nitrogen on Photosynthetic Productivity and Yield Quality of Wheat (Triticum aestivum L.).” will work.

2.     There is a lack in flow of the contents in the introduction and it’s hard to understand. I highly suggest the authors to separate the contents in Introduction section based on the factors that are explored in this study.

3.     Line 35, 38: I would highly recommend using studies and information related to wheat.

4.     Line 53: Recommend adding the full form of abbreviations like Pn and Ci (line 54) when first introduced.

5.     Line 92: Objectives need to be more coherent and clearer. Also, mention what is the hypothesis of this study?

6.     Line 102: It is not clear if the studies were conducted in controlled environment or field scenario. If controlled environment, what were the conditions maintained. If field, need more general information of year-to-year weather trend during the growing season.

7.     How many times as the study repeated?

8.     Which wheat variety was used? Spring vs winter wheat? Why this variety chosen? What is the maturity group?

9.     More information is needed on the materials and methods section. Example, how much N was already present in the soil before planting? Soil type. Etc. Which form of N was applied? Since N was already broadcasted in October how does it impact the N0 application level?

10.  Line 105: For split plot design there must be two factors. Here I see only one factor i.e., level of N application rates.

11.  Again, its unclear where the study was conducted field vs growth chamber? The study was conducted from 2019 to 2022, why only one value of CO2?

12.  Line 130: What is PEA?

13.  Statistical Analysis: More information is needed for how different years/runs of data were combined. Did you check for the requirements of ANOVA including normality of residual errors and homogeneity of variance?

14.  Line 152: Nowhere in the materials and methods Rubisco, transpiration rate is mentioned. How were they measured?

15.  Figure 1: It is very difficult to interpret the graph. Please include some simpler ways of showing the results. I am not sure what it means by bars show standard of error.

16.  Why were the results of different years not combined and pooled?

17.  Its very hard to understand the findings due to editing issues.

Reviewer 2 Report

Although this study reports on a significant and interesting theme and the results could be potentially important for wheat breeders, the manuscript has many flaws and some are listed below. In my opinion, significant changes in the description and presentation of these results and rewriting of the manuscript are required. 

- English language and style have to be corrected. Many sentences are incomplete and difficult to understand.

- 2.1.1. Photosynthetic characteristics – an instrument used for measurements should be specified.

2.1.2. Photosynthetic pigment and chlorophyll fluorescence parameters - Have you used PAM or PEA for measurements? Or maybe both? This is very hard to understand from this paragraph but should be clearly defined because there is a difference between these two types of instruments for measuring chlorophyll a fluorescence. On the other hand, the results of chlorophyll a fluorescence are not presented in your work although are indicated in the material and methods. Therefore, the description of these methods (if performed!) should be corrected and results included.

Please explain why chlorophyll fluorescence and photosynthetic characteristics were not measured in the same developmental stages (if both are indeed measured).

3.1. Rubisco activity in flag leaves of wheat after anthesis – please explain how rubisco activity was measured in your study.

Table 2. Effects of different N rate comparison of stomatal conductance gs of winter wheat.; Table 3. Effects of nitrogen application on transpiration rate E winter wheat.; Table 4. Effects of different nitrogen applications on carbon dioxide concentration Ci between different nitrogen rates in winter wheat. – you need to clearly present what was measured in your study, only measurement of PN is listed in the material and methods so please correct this and explain the parameters that have been used.